# Imitation Transformer

## Abstract

We propose a simple but effective batch imitation learning method. Our algorithm works by solving a sequence of two supervised learning problems, first learning a reward function and then using a batch reinforcement learning oracle to learn a policy. We develop a highly scalable implementation using the transformer architecture and upside-down reinforcement learning. We also analyze an idealized variant of the algorithm for the tabular case and provide a finite-data regret bound. Experiments on a set of ATARI games and MuJoCo continuous control tasks demonstrate good empirical performance.

## 1 Introduction

This work addresses batch imitation learning (BIL), an emerging setting which arises at the intersection of two system design constraints. First, in many practical scenarios, deploying live policies is expensive but we have a huge amount of preexisting and freely available exploration data that can be used for learning, motivating the use of *batch* algorithms. Second, in order to avoid unintended behavior (Russell & Norvig, 2010), it is often beneficial to train a system by specifying demonstrations rather than rewards, motivating the use of *imitation learning*.

In BIL, one thus typically assumes access to two datasets (Zolna et al., 2020; Ma et al., 2022). A small *expert* dataset contains information about the way we intend the trained agent to behave. A much larger *exploration* dataset contains information about how the world works, allowing the agent to reach expert states from an arbitrary initial configuration. A BIL algorithm is meant to compute a policy from these two datasets, with the aim of maximizing the (unknown) expert reward.

We address the BIL problem with an algorithm inspired by support matching approaches to IL (Wang et al., 2019; Reddy et al., 2019; Ciosek, 2022). They work by constructing a reward function assigning high reward to policies that match the support of the expert policy in the dataset. A policy is then learned by a single call to an RL solver with this fixed reward function. However, unlike in these previous works, we do not require online interaction with the environment in order to optimize for the constructed reward.

In the tabular setting, support estimation is straightforward. However, in order to enable more realistic settings where states can be vectors or images, we also develop a variant of our algorithm that uses a thresholded classifier to estimate support and then an Upside-Down Reinforcement Learning oracle (Schmidhuber, 2019) to learn a policy. This effectively reduces the batch IL task to solving a sequence of two supervised learning problems. We argue that this reduction-based approach, combined with the power of the transformer architecture (Chen et al., 2021), leads to increased stability of training.

**Contributions** We develop a simple, easily optimized support matching algorithm, with theoretical guarantees in the tabular Reinforcement Learning setting. Building on the work of Ciosek (2022), we provide formal bounds on the performance of a policy learned via our algorithm for discrete MDPs. These bounds depend on the size of the MDP, the size of the expert dataset, and the size and quality of the exploratory data-generating policy. As a corollary, we prove a conjecture by Kearns & Singh (1998), which to our knowledge has never been demonstrated formally until now. We go on to empirically demonstrate the effectiveness of a transformer-based relaxation of our algorithm applied in challenging MuJoCo and ATARI environments. For MuJoCo, we show that our simple algorithm is competitive in a challenging data regime

in which Behavioral Cloning (BC) fails. These results are statistically robust, making use of the evaluation methods described by Agarwal et al. (2021), which establishes a precedent for more robust benchmarking. Overall, our study finds the proposed algorithm to be both simple and effective, making it a pragmatic baseline for future work on IL in challenging, low-expert-data regimes.

## 2 Preliminaries

In this section, we provide a brief technical overview of the formalism employed in the rest of the paper and establish the notation that we will apply.

**Markov Decision Processes & Reinforcement Learning**  Markov Decision Processes (MDPs) provide a formalism for agents in sequential reasoning task. An *average reward* MDP is given by the tuple: $\langle \mathcal{S}, \mathcal{A}, P, R, P_0 \rangle$, where $\mathcal{S}$ is the state space, $\mathcal{A}$ the action space, $P : \mathcal{S} \times \mathcal{A} \to U(\mathcal{S})$ the transition distribution, where $U(\mathcal{S})$ is the set of probability distributions over states, $R : \mathcal{S} \times \mathcal{A} \to [0, 1]$ a bounded deterministic reward function, and $P_0$ the distribution over initial states. We also denote $S = |\mathcal{S}|$, $A = |\mathcal{A}|$. In this work, we consider agents which act according to a deterministic policy, $\pi : \mathcal{S} \to \mathcal{A}$, which maps states to the action taken in that state. We also assume that such policies induce *ergodic* Markov chains for simplicity. In an average reward MDP, we look to find a policy which optimizes the average per-step expected reward, given by

$$\mu_\pi = \lim_{N \to \infty} \frac{1}{N} \mathbb{E} \left[ \sum_{i=0}^{N} r\left(s_i, \pi(s_i)\right) \right],$$

where expectation is taken over draws from $P_0$ and $P\left(s_i, \pi(s_i)\right)$.

A *discounted reward* MDP adds, in addition to the tuple described above, a *discount factor*, $\gamma$, which determines an effective time horizon for the MDP. In a discounted reward MDP, values are state dependent, and given by

$$V_\pi(s) = \mathbb{E} \left[ \sum_{i=0}^{\infty} \gamma^i r\left(s_i, \pi(s_i)\right) \Big| s_0 = s \right].$$

where the conditional expectation is again taken over draws from the MDP dynamics. Related to the value function is the *state-action* value function, which determines the value of taking an action in a state, then proceeding to follow $\pi$: $Q_\pi(s, a) := r(s, a) + \gamma \mathbb{E}_{s' \sim P(s,a)} \left[ V_\pi(s') \right]$.

Of special importance to us is the fact that average rewards and the average values only differ by a constant factor of $(1 - \gamma)$ — see e.g. the work of Kakade (2001):

$$\mu^\pi = (1 - \gamma) \sum_s \rho_\pi V^\pi(s),$$

where $\rho^\pi$ is the steady-state distribution of the Markov chain induced by policy $\pi$ on the MDP.

**Imitation Learning**  Imitation learning looks to recast the problem of learning an optimal policy by making use of a fixed dataset of expert behavior, $D_E$, and by learning a policy which achieves average reward close to the one used to produce the dataset. Formally, for an expert policy, $\pi_E$ and any reward function $f$ with range $[0, 1]$, we look to learn $\pi$ such that

$$\mathbb{E}_{\rho_{\pi_E}} \left[ f(s, a) \right] - \mathbb{E}_{\rho_\pi} \left[ f(s, a) \right] \leq \epsilon. \tag{1}$$

In addition to the finite expert dataset, we consider an additional finite *exploratory* dataset, $D_X$ which consists of transitions generated by an arbitrary policy, $\pi_X$. The policy must cover the MDP, meaning it must be possible for it to visit any given state and action of the MDP in a finite number of steps. This is to say, the steady state probability obeys $\rho_\pi(s, a) \geq p_{\min} > 0$. The purpose of this dataset is to provide the offline agent with information about the MDP dynamics outside of the states covered by the expert policy. Training thus occurs on the unified dataset $D_U = D_E \cup D_X$.

---

**Algorithm 1** Imitation Learning by Batch RL

---

**Input:** Expert dataset: $D_E$, Exploration dataset: $D_X$
**Output:** Imitation policy: $\pi$
 1: $D_U \leftarrow D_E \cup D_X$
 2: **for** $s, a \in D_U$ **do**
 3:     $\hat{r}(s, a) \leftarrow \mathbf{1}_{D_E}((s, a))$
 4: **end for**
 5: $\pi \leftarrow \text{BatchRL}(D_U, \hat{r})$
 6: **return** $\pi$

---

**Total Variation Distance**   The total variation distance (TV) formalizes a notion of how far apart two probability distributions are, in terms of how they assign different probabilities to the same events. Formally, for discrete distributions $\rho_1$ and $\rho_2$, over an event set $\Omega$, we have

$$\|\rho_1 - \rho_2\|_{TV} := \sup_{M \subseteq \Omega} |\rho_1(M) - \rho_2(M)|.$$

It is well-known that bounding the total variation distance between the expert policy and a policy learned via imitation learning also provides a bound on Equation (1) — see, e.g. Ciosek (2022). As such our bound will also look to bound the TV between the expert distribution and the policy learned by our algorithm. The total variation distance is also used to define the *mixing time* $t^\pi$ of an ergodic policy $\pi$, as the smallest $t$ such that

$$\max_s \|\mathbf{1}(s)^\top P_\pi^t - \rho_\pi\|_{TV} \le \frac{1}{4},$$

where $P_\pi$ is the transition matrix for the Markov chain induced by $\pi$ and transition function $P$ and $\mathbf{1}(s)$ is an indicator vector with one in position corresponding to state $s$. This will be used in our analysis to represent the quality of the exploratory policy, and the coverage of the expert data over the state space.

## 3   Imitation Learning by Batch RL

In order to introduce our algorithm we define the *intrinsic* reward, which, for discrete MDPs, is given by

$$\hat{r}(s, a) = \mathbf{1}_{D_E}((s, a)),$$

where $\mathbf{1}_{D_E}$ is the indicator function, which maps to 1 if a state action pair is in the expert dataset, and 0 otherwise. This matches exactly the function used by Wang et al. (2019) and Ciosek (2022). However, we implement a subtle but important shift. Instead of making a call to an arbitrary RL solver in order to optimize returns under this intrinsic reward through unlimited interaction with the environment, we make a call to an *offline* RL algorithm, which in general may not be able to perfectly optimize for our intrinsic reward. This significantly complicates analysis, as we need to manage the inefficiency and statistical dependence on finite data that comes from offline RL algorithms, alongside the fact that optimizing for the intrinsic reward may not lead to a good policy if there is not enough expert data. Pseudo-code for the overall process is given in Algorithm 1.

In order to simplify analysis, we make use of an off-policy RL algorithm, developed by Kearns & Singh (1998), known as *phased Q-Learning*. The algorithm is somewhat impractical, but its analysis is tractable. We develop favorable asymptotic guarantees on its sample complexity in subsequent sections.

Later on, in section 6, we use the transformer architecture to extend this algorithm to the setting where the state-action space is non-tabular.

## 4   Prior Work

**Behavioral Cloning**   There exists a wide body of prior work which looks to solve the imitation learning problem. The simplest and oldest form is known as behavioral cloning, which simply optimizes policy

parameters in order to match the expected state-conditional probability of taking the actions in the expert dataset. However, this formulation fails to accommodate for the sequential nature of the MDP, leading to distribution shift (Ross & Bagnell, 2010), and severely increasing the amount of expert data needed to learn a good policy.

**Inverse RL**   Inverse RL (IRL) proposes to evade this issue by learning a reward function which the expert is (implicitly) maximizing — as indicated by the data — then optimizing a policy to solve the original MDP augmented with this reward function. Early formulations of this problem required full solutions to candidate MDPs at each iteration (Ng et al., 2000; Ziebart et al., 2008), while more recent *adversarial* IRL methods (Ho & Ermon, 2016) use GAN-style training to construct candidate reward functions and policies online in order to minimize a divergence between the learned policy and the expert data distribution. However, training of these methods is known to be difficult to stabilize. Also, they require continuous interaction with the environment, which is not available in our setting, which constrains interaction with the MDP to a finite dataset.

**Support Matching**   In order to avoid the instabilities of adversarial IRL methods, and to limit the number of calls to an RL solver needed, Wang et al. (2019) propose a phased approach, which looks to construct a reward function such that state-actions visited by (or close to those visited by) a deterministic expert are rewarding, while state-actions far from the dataset are not. A reward function is first estimated using only the expert data, either by direct labeling, or an approximate support estimation method. Then, a single call to an RL solver is made, using the rewards supported by the expert. Ciosek (2022) develops theory for support matching methods, providing formal guarantees for reward labeling on discrete MDPs, as well as a heuristic relaxation of the labeling which has been shown to work in continuous environments. Our work builds on these support matching results, but looks to do so in a strictly batch setting, without the need to call an online RL solver.

**Offline Imitation Learning**   Several previous methods employ the setting assumed by our work, looking to learn to mimic an expert policy strictly from data. Zolna et al. (2020) propose ORIL, which is a classifier-based approach. It is similar to our work but crucially differs in that the rewards are not binary, instead being proportional to the probability of classifying the state-action pair as coming from the expert. ORIL-style algorithms do not have a theoretical guarantee on the performance of the resulting policy. Moreover, we will see in section 7 that ORIL can fail even for simple grid-worlds. Another, related approach is to consider an offline version of SQIL (Reddy et al., 2019), which does have binary rewards but where the *expected* reward is the same as for ORIL. It suffers form the same shortcomings. More recently, Kim et al. (2021) establish the use of a hybrid dataset composed of expert and exploratory data, making use of a marginalized density ratio method (Liu et al., 2018; Nachum et al., 2019) in order to minimize the KL-divergence between the learned policy and expert behavior. Similarly, Ma et al. (2022) discuss a paradigm which leverages a more general class of $f$-divergences.

**Transformers and Decision Transformers**   Transformers were introduced by Vaswani et al. (2017) in the context of large language models. The transformer architecture provides a way to learn sequence-to-sequence models which are amenable to be easily trained with back-propagation. Radford et al. (2019) proposed GPT2, a particularly effective transformer model, on a variant of which our technique is based. Decision Transformers (Chen et al., 2021) extend this idea to batch reinforcement learning, by casting it as a sequence modeling task.

**Upside-Down Reinforcement Learning**   Schmidhuber (2019) proposed Upside Down Reinforcement Learning as a way of battling the instability inherent in Deep Reinforcement Learning methods. Instead of learning critic functions, which map state-action pairs to values, Upside Down RL learns a mapping from states and 'commands' to actions, which can be done using supervised learning. This paper uses a batch RL oracle which is based on the simplest form of Upside Down RL, where the commands correspond to the desired (intrinsic) return of the agent. Since we are in the batch setting, we avoid the difficulties inherent in integrating online exploration (Srivastava et al., 2019) with Upside Down RL.

# 5 Theory for the Tabular Case

In this section, we provide a proof for a regret bound for the algorithm. Readers more interested in practical applications might want to skip this section and proceed directly to the experiments.

Our proof is structured in the following way. First, in section 5.1, we study the properties of our chosen RL oracle (phased Q-learning). Then, in section 5.2, we use that result combined with an adaptation of the techniques of Ciosek (2022) (which were originally developed for online RL) to show a regret bound.

## 5.1 Theory of Phased Q-Learning

Phased Q-learning can be seen as an approximate form of value iteration. In this algorithm, rather than sampling states and actions individually, each action at each state across the entire environment is sampled simultaneously $m$ times. This data generation process is known as *parallel sampling*. Parallel sampling can be simulated using standard RL sampling techniques based on rollouts of a policy in the MDP, which we characterize theoretically later in this section. It allows us to update the estimated state-action value at each state and action simultaneously according to the optimal Bellman equation using the deterministic reward and *empirical bootstrap*:

$$Q_{i+1}(s,a) = r(s,a) + \gamma \frac{1}{m} \sum_{k=1}^{m} \max_{a'} Q_i(s'_k, a').$$

Since the empirical bootstrap will concentrate around the true value, we can expect that, with infinite data, this becomes equivalent to performing value iteration. Here we prove that this is an efficient offline RL algorithm. First, we need to demonstrate that the amount of data needed to achieve bounded error at the end of the evaluation process is polynomial in the appropriate quantities, which was accomplished in the original work by Kearns & Singh (1998). Second, we need to show that we can simulate sampling from the parallel model efficiently. Kearns & Singh (1998) postulate that such a procedure should be possible under assumptions concerning the quality of the exploration policy, but a formal proof has never been provided until now.

We use the notation from Kearns & Singh (1998). $m$ is the number of samples from the parallel sampler per state-action pair, $\ell$ is the number of iterations of phased Q-learning to perform, $s$ and $a$ are states and actions, and $P_{ss'}^a$ is the transition distribution, $r$ is a deterministic reward function, $\hat{V}_i(\cdot), \hat{Q}_i(\cdot, \cdot)$ are our estimated value functions at iteration $i$, $\bar{V}_i(\cdot), \bar{Q}_i(\cdot, \cdot)$ are the true value functions of the policy greedy with respect to $\hat{Q}_i$, $V_i(\cdot), Q_i(\cdot, \cdot)$ is the output of $i$ rounds of policy iteration, $V^*(\cdot), Q^*(\cdot, \cdot)$ are the optimal value functions.

We start by assuming that the empirical bootstrap of our estimated function is $\eta'$-concentrated around the bootstrap on the true transition distribution. We will later take $m$ large enough for this to hold. For all $s, a$, and $i \leq \ell$, we have

$$\left| \frac{1}{m} \sum_{k=1}^{m} \hat{V}_i(s'_k) - \sum_{s'} P_{ss'}^a \hat{V}_i(s') \right| \leq \eta', \tag{2}$$

where $s'_k$ is the $k$th sampled subsequent state from the parallel sampler at $s, a$.

With this assumption, we are able to formalize the notion that performing phased Q-learning is very close to performing true value iteration, as suggested by the following lemma.

**Lemma 1.** *Given Equation* (2)*, the difference between the value output by phased Q-learning after $\ell$ steps and the optimal value is given by*

$$\zeta(s,a) := \left| \hat{Q}_\ell(s,a) - Q^*(s,a) \right| \leq \frac{\eta' \gamma + \gamma^\ell}{1 - \gamma} \qquad \forall s, a.$$

The proof is left to the appendix, which makes use a recurrence relation between updates of phased Q-learning.

We now look to bound the overall suboptimality of the policy output by phased Q-learning. We define the discounted regret of the policy at step $i$ on the discounted MDP as the expectation of the difference between

the discounted return achieved by the learned policy and that of the optimal policy, with expectation taken over the initial state distribution:

$$R_\gamma(i) := \mathbb{E}_{s_0} \left[ V^*(s_0) - \bar{V}_i(s_0) \right].$$

Equipped with the pointwise value gap, we can bound the overall regret with the following lemma.

**Lemma 2.** *Given Equation* (2)*, the policy output by phased Q-learning after $\ell$ steps achieves regret of at most*

$$R_\gamma(\ell) \leq \frac{2}{(1-\gamma)} \left( \frac{\eta'\gamma + \gamma^\ell}{1-\gamma} \right).$$

We provide a complete proof of Lemma 2 in the appendix. Since we now have a bound on the overall regret of our algorithm, bounding by $\epsilon$, our overall error margin, allows us to solve for the relevant quantities, eventually leading to a lower bound on the number of samples needed to achieve such error.

Once we have solved for the minimum acceptable concentration error $\eta'$ in terms of our overall allowable discounted regret, $\eta$, we can solve for $m$, the number of samples we need for Equation (2) to hold with high probability. This is accomplished in the following lemma.

**Lemma 3.** *Taking $m\ell$ samples from the parallel model for phased Q-learning, such that*

$$m\ell \geq \log\left( \frac{2\ell S A}{\delta'} \right) \frac{2\gamma^2 \ell}{(1-\gamma)^2 (\eta(1-\gamma)^2 - 2\gamma^\ell)^2},$$

*we obtain discounted regret less than $\eta$ with probability at least $1 - \delta'$.*

The proof is left to the Appendix. It involves bounding by the overall error $\eta$ and solving for $m\ell$ using the Hoeffding inequality.

Now we only need to concern ourselves with the gathering of these $m\ell$ samples from the parallel model. We do so by appealing to a classic result applying to ergodic Markov chains, which suggests that sampling the current state and action after following the chain for several steps is close to sampling from the stationary distribution, allowing for gathering of independent samples.

**Lemma 4.** *The number of sample transitions from the exploration policy needed to perfectly simulate the parallel sampling model with failure probability less than $\delta''$ is upper bounded by*

$$\frac{2t^{\pi_X}}{\log(2)p_{\min}} \log\left( \frac{2}{p_{\min}} \right) \log\left( \frac{SA}{\delta''} \right).$$

The proof can be found in the Appendix. Lemma 4 combined with Lemma 3 give the overall sample complexity of the phased-Q learning stage of our algorithm.

## 5.2 Bounding the TV Between Expert and Imitator

With a good sense for the data requirements of our offline RL algorithm, we can proceed to ensure performance of our algorithm. We do so by invoking a key lemma of Ciosek (2022). However, this lemma applies only to the average reward setting, while our results thus far concern policies optimized for discounted reward. As such we must first bound the suboptimality of the learned policy with respect to the average value attained. We achieve this with a relaxation of Theorem 1 by Kakade (2001).

**Lemma 5.** *Suppose policy $\pi$ is $\epsilon$-suboptimal or better at each state with respect to the discounted return. Let $\pi^*$ be the optimal policy under the average reward criterion. We assume $P_{\pi^*}$ has $S$ distinct eigenvalues. Taking $\Sigma$ to be the matrix of right eigenvectors of $P_{\pi^*}$ with corresponding eigenvalues: $\lambda_1 = 1 > ... \geq |\lambda_S|$, we have*

$$\mu^\pi \geq \mu^{\pi^*} - \kappa(\Sigma)\|\mathbf{r}\| \frac{1-\gamma}{1-\gamma|\lambda_2|} - (1-\gamma)\epsilon,$$

*where $\kappa(\Sigma) := \|\Sigma\|_2 \|\Sigma^{-1}\|_2$ is the condition number of the matrix $\Sigma$, and $\mathbf{r}$ is the vector with entries $r(s, \pi^*(s))$ for each state.*

The proof is in the appendix. It follows closely that of Kakade (2001).

Lemma 5 gives us a lower bound on the suboptimality with respect to the average reward of a policy $\epsilon$-suboptimal with respect to the discounted reward. Next, we need to ensure that it is possible to achieve a policy which performs well with respect to the intrinsic reward. This is accomplished by invoking Lemma 5 of Ciosek (2022), reproduced here for convenience.

**Lemma 6** (Lemma 5, Ciosek, 2022)**.** *Given an expert dataset consisting of $|D_E|$ points, a policy $\pi$ which maximises the average intrinsic reward achieves an expected average intrinsic reward of at least*

$$\mu_{int}^{\pi} := 1 - \nu - \sqrt{\frac{8St^{\pi_E}}{|D_E|}},$$

*for all error terms $\nu > 0$, with probability at least*

$$1 - \delta''' := 1 - 2\exp\left(-\frac{\nu^2|D_E|}{4.5t^{\pi_E}}\right).$$

To connect high performance on the intrinsic reward with the performance on the extrinsic reward, we again invoke a key lemma of Ciosek (2022),

**Lemma 7** (Lemma 7, Ciosek, 2022)**.** *An agent which achieves an average reward of $1 - \epsilon'$ on the intrinsic reward MDP also achieves average reward of*

$$(1 - \epsilon')\mu^{\pi_E} - 4t^{\pi_E}\epsilon'$$

*on the true MDP.*

We are now ready to conclude, evaluating the overall sample complexity of our algorithm.

**Theorem 1.** *Consider an instance of Algorithm 1. In our setting (Assumptions 1-4 in the appendix), in order to achieve regret on the average reward problem of less than $\epsilon$, or total variation distance between the expert and the imitation policy less than $\epsilon$, with probability $1 - \delta$, we need to sample*

$$\max\left\{\frac{1}{\epsilon^2}32St^{\pi_E}(1 + 4t^{\pi_E})^2,\ \frac{4.5t^{\pi_E}16(1 + 4t^{\pi_E})^2\log(4/\delta)}{\epsilon^2}\right\}$$

*transitions from the expert policy, and*

$$\mathcal{O}\left(\frac{t^{\pi_X}}{p_{\min}}\log\left(\frac{1}{p_{\min}}\right)\log^2\left(\frac{SA}{\delta}\right)\log\left(\frac{t^{\pi_E}\beta}{\epsilon(1-\lambda)}\right)\frac{(t^{\pi_E})^8\beta^6}{\epsilon^8(1-\lambda)^6}\right)$$

*from the exploratory policy, where $\beta = \kappa(\Sigma)\|\mathbf{r}\|$, as described in Lemma 5.*

We defer the proof to the Appendix, which largely consists of disentangling the relevant constants. We note that the bound on the amount of expert data needed differs from that of Ciosek (2022) only in terms of a constant factor. This indicates that the burden of offline learning falls mostly on the exploratory policy. We use big-O notation to denote the number of exploratory samples needed in order to provide intuition, but specific constants can be found in the proof of Theorem 1.

## 6    Imitation Transformer and the Extension to Non-Tabular State-Action Spaces

Our reward construction crucially depends on identifying state-action pairs that belong to the expert dataset. The algorithm defined in the previous section used a tabular state-action space, where the question of whether one state-action pair matches another can be resolved trivially.

On the other hand, one would want to use batch imitation learning in non-tabular settings where states can be images or arbitrary vectors. We now present the imitation transformer architecture, which targets this use-case. The pseudo-code is given in Algorithm 2.

The algorithm works in three stages. First, we train a classifier that distinguishes expert state-action pairs from exploration state-action pairs. We then threshold the classifier outputs to obtain a reward signal. Finally, we call a decision transformer with the obtained binary reward.

---

**Algorithm 2** Imitation Transformer

---

**Input:** Expert dataset: $D_E$, Exploration dataset: $D_X$, threshold $\tau$
**Output:** Imitation policy: $\pi$
 1: $D_U \leftarrow D_E \cup D_X$
 2: $c \leftarrow \text{trainClassifier}(D_E, D_X)$
 3: **for** $s, a \in D_U$ **do**
 4: $\quad \hat{r}(s, a) \leftarrow \mathbf{1}_{D_E}(c(s, a) \geq \tau)$
 5: **end for**
 6: $\pi \leftarrow \text{DecisionTransformer}(D_U, \hat{r})$
 7: **return** $\pi$

---

| Algorithm | First Exploration Dataset | Second Exploration Dataset |
|---|---|---|
| IT | **0.7104** $\pm$ 0.0016 | **0.7326** $\pm$ 0.0012 |
| BC (expert) | 0.4873 $\pm$ 0.0035 | 0.4873 $\pm$ 0.0035 |
| BC (explore) | 0.0786 $\pm$ 0.0020 | 0.1635 $\pm$ 0.0029 |
| BC (mixed) | 0.0845 $\pm$ 0.0020 | 0.1874 $\pm$ 0.0026 |
| DWBC | 0.4803 $\pm$ 0.0029 | 0.4809 $\pm$ 0.0032 |
| DemoDICE | 0.6184 $\pm$ 0.0025 | 0.6226 $\pm$ 0.0031 |
| FR/SQIL | 0.6651 $\pm$ 0.0015 | 0.0000 $\pm$ 0.0000 |
| ORIL-BASIC | **0.7101** $\pm$ 0.0015 | 0.0000 $\pm$ 0.0000 |
| ORIL-SPLIT | 0.7034 $\pm$ 0.0035 | 0.0000 $\pm$ 0.0000 |
| (Expert) | 0.7407 | 0.7407 |

Table 1: Mean performances on environment during evaluation for 50 training seeds, averaged over 100 rollouts. Error measures are given by a 95% confidence interval based on the mean performance per training seed.

# 7 Experiments

## 7.1 Discrete Gridworld

We begin by performing an empirical evaluation of our algorithm on a discrete gridworld (see appendix B for task details). This relatively simple setting allows us to strictly control all aspects of the algorithm. The main aim of this experiment is to pick the most promising algorithms to test against when we compare against the transformer-based implementation in the subsequent sections.

Results are given Table 1. We report results on three variants of classic behavioral cloning, trained on the expert dataset, the exploration dataset and a concatenation of the two datasets respectively. We also tried discriminator-weighted behavioral cloning (DWBC), a new method by Xu et al. (2022) that computes weights for the behavioral cloning step adaptively. In addition, we implemented two variants of ORIL. ORIL-BASIC learns a classifier on the whole dataset, while ORIL-SPLIT uses a subset as per Zolna et al. (2020). We also compare to an offline version of SQIL[1] (Reddy et al., 2019), a noisy version of a classifier-based algorithm. Moreover, compare with DemoDICE (Kim et al., 2021), an approach minimizing the KL-divergence between the expert and the imitator.

Our experiments demonstrate conclusively that the Imitation Transformer ties with ORIL for best performance on the first exploration dataset and outperforms all competing methods on the second. This confirms two important intuitions. First, behavioral cloning methods, even with a complex weight tuning mechanism, fail on tasks which require planning many steps ahead. Second, ORIL fails to learn on certain exploration datasets, despite the fact that the second exploration dataset is much larger than the first. This could be expected given ORIL has no formal guarantees on the quality of the resulting policy.

---

[1]We denote this algorithm FR/SQIL since Zolna et al. (2020) refer to it as 'flat rewards'.

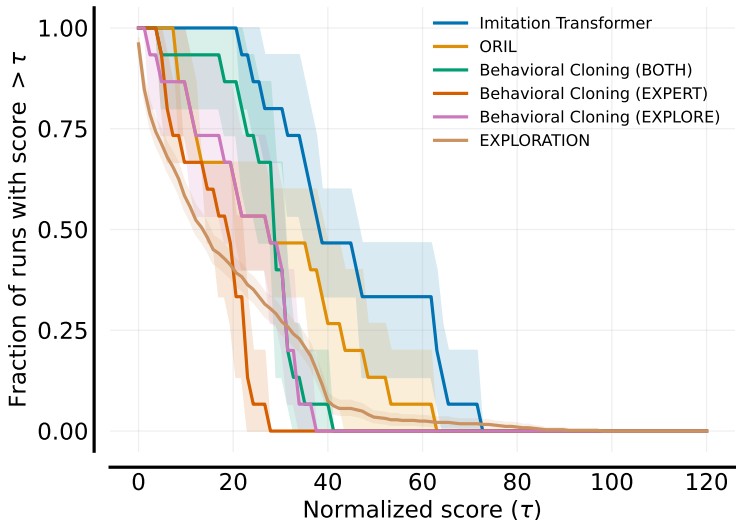

Figure 1: Performance profiles for Imitation Transformer and baselines (Agarwal et al., 2021) for MuJoCo. Curves that arc further from the origin are higher performing. The Imitation Transformer is competitive against both ORIL and behavioral cloning

## 7.2 Imitation Transformer Experiments on Continuous Control Tasks

In order to evaluate the efficacy of IT in a more complex scenario, we perform experiments in the MuJoCo (Brockman et al., 2016) environment. We selected ORIL as the main baseline we are going to compare to, mainly due to its good performance on the gridworld task as well as due to the fact that it can be seamlessly integrated with the transformer architecture. We also compare to several versions of Behavioral Cloning, as well as the performance of the exploration policy.

In our experiments, data comes from the D4RL MuJoCo benchmarks (Fu et al., 2020), which are standard continuous control datasets used to benchmark state-of-the-art offline RL and IL methods. The D4RL data consists of several datasets, generated by policies of varying quality. The "expert" data represents rollouts from a stochastic agent trained on the true environment using SAC — this policy is taken to be close to optimal in the environment. We use a fraction of the D4RL expert data as our expert dataset. In order to simulate the use of known safe exploratory policies for training and evaluation, our exploratory data, is derived from the D4RL "medium-replay" dataset, which comes from the replay buffer of a suboptimal, intermediate agent trained on the true environment.

Our implementation of all the algorithms was based on the transformer architecture[2], where we solve the batch reinforcement learning using the decision transformer (Schmidhuber, 2019; Chen et al., 2021). The classifier used to distinguish expert from non-expert demonstrations is also based on the same transformer architecture. We provide further details about our implementation in appendix C.

Results are shown in Figure 1. It can be seen that the Imitation Transformer achieves the best overall performance. The best version of Behavioral Cloning turns out to be the one with the largest dataset, which includes a combination of expert and exploration data. Overall, Behavioral Cloning ties with ORIL for the second place, showing that ORIL was doing well for a small fraction of tasks, while BC was doing slightly worse, but for a larger fraction of tasks. As a sanity check, we also confirm that the Imitation Transformer, ORIL and the best version of Behavioral Cloning perform better than the exploration dataset, showing learning is definitely taking place. We show more ablations in appendix C.

---

[2]We used https://github.com/karpathy/nanoGPT as the basis of our implementation.

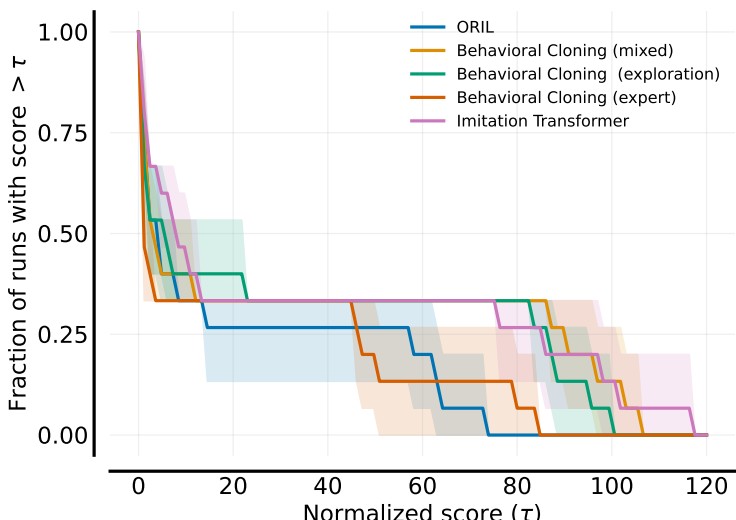

Figure 2: Performance profiles for Imitation Transformer and baselines (Agarwal et al., 2021) for ATARI games. Curves that arc further from the origin are higher performing.

### 7.3 Imitation Transformer Experiments on Atari

In order to provide further evaluation of our algorithm, we also tried it on 3 ATARI games from the Arcade Learning Environment (Bellemare et al., 2013): Pong, Qbert and Seaquest. Results are shown in Figure 2, which summarizes results from 15 experiments (5 runs per game).

It can be seen that the Imitation Transformer outperforms ORIL, confirming our intuition that thresholding the reward function is crucial for good performance. Somewhat disappointingly, on this domain, IT has a statistically comparable performance to the variants of behavioral cloning trained on exploration data and a mixture of expert and exploration data.

Further details about the ATARI experiments as well as hyperparameters are available in Appendix D.

## 8 Conclusions

We have developed and a simple and effective algorithm for imitation learning from batch data. We provided a theoretical analysis of the algorithm for the tabular case. We have also provided a scalable implementation based on the transformer architecture.

### Broader Impact Statement

Ours is a methods paper which uses synthetic benchmarks. The ethical implications of our work are the same as for most attempts to improve batch RL or imitation learning. While the methods can surely be put to nefarious uses, we believe that the benefits offered by new, more efficient RL outweigh such risks. In addition, there is a risk of over-reliance on our theoretical bound in cases where the assumptions are not fully met. To help alleviate this risk, we have clearly identified all assumptions in the appendix.

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

## A   Proofs

In this section we provide detailed proof of the results that appear in the main paper. We also extend Theorem 1 to include a bound on total variation distance between the expert and imitation policy, which follows directly from the regret bound with the same constant. We recall the setting of our work through the following key assumptions.

*Assumption* 1. The expert and imitation policies are deterministic.

*Assumption* 2. Expert, exploration, and imitation policies induce aperiodic and irreducible Markov chains.

*Assumption* 3. The Markov chain induced by the exploratory policy has a uniform lower bound on the probability that the agent will visit any given state and action:

$$\rho_{\pi_X}(s, a) > p_{\min}.$$

*Assumption* 4. The transition matrix obtained by following policy $\pi^*$ on the MDP, $P_{\pi^*}$ has $n$ distinct eigenvalues.

Assumption 1 cannot be relaxed as it is impossible to imitate a stochastic expert with a single call to the batch RL algorithm with a stationary reward function-optimisation may in general lead to a deterministic policy, as any stationary MDP has a deterministic optimal policy. Assumption 2 can, for the expert and imitation policies, possibly be relaxed to the periodic setting with appropriate attention to detail. However, in order to achieve appropriate sampling over the state space, this cannot be relaxed for the exploratory policy. Similar concerns suggest that Assumption 3 cannot be relaxed without introducing other assumptions. Assumption 4 is only needed to prove Lemma 5, and can, in principle, be relaxed.

### A.1   Theory of Phased Q-Learning

We use the notation from Kearns & Singh (1998):

- $m$ is the number of samples from the parallel sampler,

- $\ell$ is the number of iterations of phased Q-learning,

- $s$ and $a$ are states and actions, and $P_{ss'}^a$ is the transition distribution,

- $r$ is a deterministic reward function,

- $\hat{V}_i(\cdot), \hat{Q}_i(\cdot, \cdot)$ are our estimated value functions at iteration $i$,

- $\bar{V}_i(\cdot), \bar{Q}_i(\cdot, \cdot)$ are the true value functions of the policy greedy with respect to $\hat{Q}_i$,

- $V_i(\cdot), Q_i(\cdot, \cdot)$ is the output of $i$ rounds of policy iteration,

- $V^*(\cdot), Q^*(\cdot, \cdot)$ are the optimal value functions.

In addition, we use the following notation for different error variables, reproduced here for reference:

| Error Variable | Interpretation |
|:---:|:---:|
| $\epsilon$ | Output error of Algorithm 1 |
| $1 - \epsilon'$ | Average intrinsic reward obtained by the learned policy |
| $\epsilon''$ | Reward construction error |
| $1 - \nu$ | Probability controllable intrinsic reward of expert policy. |
| $\eta$ | Discounted regret for phased Q-learning |
| $\eta'$ | Concentration error of sampled values |

Table 2: Error Variables

Following the intuition that phased Q-Learning is similar to value iteration, we make use of the following inequality–that our estimate of the bootstrapped value concentrates around the true bootstrap:

**Claim 1.** *For all $s, a$, and $i \leq \ell$, we have*

$$\left| \frac{1}{m} \sum_{k=1}^{m} \hat{V}_i(s_k') - \sum_{s'} P_{ss'}^a \hat{V}_i(s') \right| \leq \eta', \tag{3}$$

*where $s_k'$ is the kth sampled subsequent state from the parallel sampler at $s, a$.*

We will later take $m$ large enough to demonstrate that Claim 1 will hold with high probability. This is done in the proof of Lemma 3.

### A.1.1 Proof Of Lemma 1

**Lemma 1.** *Given Equation (3), the difference between the value output by phased Q-learning after $\ell$ steps and the optimal value is given by*

$$\zeta(s, a) := \left| \hat{Q}_\ell(s, a) - Q^*(s, a) \right| \leq \frac{\eta' \gamma + \gamma^\ell}{1 - \gamma} \qquad \forall s, a.$$

*Proof.* We begin by bounding the gap between value estimated by phased-Q iteration and the value output by policy iteration. Recall the phased Q-learning update, given by

$$\hat{Q}_{i+1}(s, a) = r(s, a) + \gamma \frac{1}{m} \sum_{k=1}^{m} \hat{V}_i(s_k').$$

For any $s, a$ pair, this means we have

$$
\begin{aligned}
\Big| \hat{Q}_{i+1}&(s,a) - Q_{i+1}(s,a) \Big| \\
&= \Big| r(s,a) + \gamma \frac{1}{m} \sum_{k=1}^{m} \hat{V}_i(s'_k) - r(s,a) - \gamma \sum_{s'} P^a_{ss'} V_i(s') \Big| \\
&\leq \gamma \Big| \sum_{s'} P^a_{ss'} \left( \hat{V}_i(s') - V_i(s') \right) \Big| + \gamma \eta' \\
&\leq \gamma \max_{s'} \Big| \hat{V}_i(s') - V_i(s') \Big| + \gamma \eta' \\
&\leq \gamma \max_{s'} \Big| \max_{a'} \hat{Q}_i(s',a') - \max_{a'} Q_i(s',a') \Big| + \gamma \eta' \\
&\leq \gamma \max_{s',a'} \Big| \hat{Q}_i(s',a') - Q_i(s',a') \Big| + \gamma \eta',
\end{aligned}
$$

where the first inequality makes use of our claim in (3), the second uses the fact that we have a convex combination of the value gap over states, and the last from the max of a difference being greater than a difference of max. Starting from $Q_0(s,a) = \hat{Q}_0(s,a)$ and recursively applying this equation leads to the bound

$$
\Big| \hat{Q}_\ell(s,a) - Q_\ell(s,a) \Big| \leq \frac{\eta' \gamma}{1 - \gamma}, \tag{4}
$$

by the geometric series and the fact that both algorithms are initialised identically.

Puterman (2005, Thm 6.3.3) shows that $|Q_\ell(s,a) - Q^*(s,a)| \leq \gamma^\ell / (1 - \gamma)$. Combining with the above yields

$$
\Big| \hat{Q}_\ell(s,a) - Q^*(s,a) \Big| \leq \frac{\eta' \gamma + \gamma^\ell}{1 - \gamma}. \qquad \square
$$

### A.1.2 Proof of Lemma 2

Recall that the discounted regret is defined as

$$
R_\gamma(i) := \mathbb{E}_{s_0} \left[ \bar{V}^*(s_0) - \bar{V}_i(s_0) \right].
$$

**Lemma 2.** *Given Equation (3), the policy output by phased Q-learning after $l$ steps achieves discounted regret of at most:*

$$
R_\gamma(\ell) \leq \frac{2}{(1 - \gamma)} \left( \frac{\eta' \gamma + \gamma^l}{1 - \gamma} \right).
$$

*Proof.* We begin by bounding the statewise suboptimality of the learned policy using the suboptimality of a single action. Let $\pi_\ell(s) := \operatorname{argmax}_a \hat{Q}_\ell(s,a)$ give the policy output after $\ell$ steps of phased Q-learning. We abuse notation slightly here by writing $a := \pi_\ell(s)$, since context is clear. Then we have

$$
\begin{aligned}
\Big| \bar{V}_\ell(s) - V^*(s) \Big| &= \Big| r(s,a) + \gamma \sum_{s'} P^a_{ss'} \bar{V}_\ell(s') - V^*(s) \Big| \\
&= \Big| \gamma \sum_{s'} P^a_{ss'} [\bar{V}_\ell(s') - V^*(s')] + Q^*(s,a) - V^*(s) \Big| \\
&\leq \gamma \Big| \max_{s'} [\bar{V}_\ell(s') - V^*(s')] \Big| + \Big| Q^*(s,a) - V^*(s) \Big|.
\end{aligned}
$$

Unrolling this gives rise to a geometric series in $\gamma$:

$$
|\bar{V}_\ell(s) - V^*(s)| \leq \frac{1}{1 - \gamma} \max_s |Q^*(s, \pi_\ell(s)) - V^*(s)|. \tag{5}
$$

Let $\pi^*(s) := \arg\max_a Q^*(s, a)$ be the optimal policy. We bound this gap for $a = \pi_\ell(s) \neq \pi^*(s) = a^*$, since otherwise it is zero. Since, by construction, $\hat{Q}_\ell(s, a^*) \leq \hat{Q}_\ell(s, a)$, the gap is maximised when we are both underestimating the optimal value of the optimal action and overestimating the optimal value of a suboptimal action:

$$
\begin{aligned}
\left|Q^*(s, a) - V^*(s)\right| &= Q^*(s, a^*) - Q^*(s, a), \\
&= Q^*(s, a^*) - \hat{Q}_\ell(s, a^*) + \hat{Q}_\ell(s, a^*) - \hat{Q}_\ell(s, a) + \hat{Q}_\ell(s, a) - Q^*(s, a), \\
&\leq |Q^*(s, a^*) - \hat{Q}_\ell(s, a^*)| + \hat{Q}_\ell(s, a^*) - \hat{Q}_\ell(s, a) + |\hat{Q}_\ell(s, a) - Q^*(s, a)|, \\
&\leq 2\left(\frac{\eta'\gamma + \gamma^\ell}{1 - \gamma}\right) + \underbrace{\hat{Q}_\ell(s, a^*) - \hat{Q}_\ell(s, a)}_{\leq 0},
\end{aligned}
$$

where the final line is an application of Lemma 1. Since the both this and Equation (5) apply across all states, combining this with Equation (5) gives the bound on our regret. □

### A.1.3 Proof of Lemma 3

We recall Hoeffding's Inequality:

**Lemma** (Hoeffding's Inequality). *Let $X_j$ be a sequence of i.i.d. random variables uniformly bounded by $0 \leq X_j \leq a$. Define the sample mean of the sequence of length $m$ as $\bar{X}_m := \frac{1}{m}\sum_{j=1}^m X_j$. Then we have*

$$
P\left(\left|\bar{X}_m - \mathbb{E}[X_1]\right| \geq t\right) \leq 2\exp\left(-2\frac{t^2 m}{a^2}\right).
$$

Using Hoeffding's Inequality, once we have solved for the minimum acceptable concentration error $\epsilon'$, we can solve for $m$, the number of samples we need for Equation (3) to hold with high probability. This is accomplished in the following lemma.

**Lemma 3.** *Taking $m\ell$ samples from the parallel model for phased Q-learning, such that*

$$
m\ell \geq \log\left(\frac{2\ell SA}{\delta'}\right)\frac{2\gamma^2\ell}{(1 - \gamma)^2(\eta(1 - \gamma)^2 - 2\gamma^l)^2},
$$

*we obtain discounted regret less than $\eta$ with probability at least $1 - \delta'$.*

*Proof.* In order to bound the overall error by $\eta$, we choose the phased Q-learning concentration error as

$$
\eta' = \left[\eta(1 - \gamma)^2/2 - \gamma^\ell\right]/\gamma,
$$

with the additional condition that:

$$
\ell \geq \frac{\log\eta + 2\log(1 - \gamma) - \log 2}{\log\gamma}, \tag{6}
$$

to ensure that $\eta'$ is not negative.

Substituting our choice of $\eta'$ into Lemma 2, we see that regret obeys

$$
R_\gamma(\ell) \leq \frac{2}{(1 - \gamma)}\left(\frac{\eta'\gamma + \gamma^l}{1 - \gamma}\right) = \frac{2}{(1 - \gamma)}\left(\frac{\frac{\eta(1-\gamma)^2/2 - \gamma^\ell}{\gamma}\gamma + \gamma^l}{1 - \gamma}\right) = \eta.
$$

With our error bounded, we proceed to bound the probability with which we fail to achieve this error, which will lead to our overall bound on $m$. We define the concentration error as

$$
\Delta_i(s, a) := \left|\frac{1}{m}\sum_{k=1}^m \hat{V}_i(s'_k) - \sum_{s'} P^a_{ss'}\hat{V}_i(s')\right|.
$$

Next, we apply Hoeffding's Inequality for each $s, a, i$, substituting $X_j = \hat{V}_i(s'_k)$ and $t = \eta'$. At step $i$, conditioned on $s$ and $a$, $\hat{V}_i$ is a deterministic function with range $[0, (1-\gamma)^{-1}]$ and the $s'_k$ are independent draws from the MDP transition function at $s, a$, so we obtain

$$P\left(\Delta_i(s, a) \geq \eta'\right) \leq 2 \exp\left\{-\frac{2m\left([\eta(1-\gamma)^2/2 - \gamma^\ell]/\gamma\right)^2}{(1-\gamma)^{-2}}\right\}.$$

We take a union bound over state, action, and algorithm step, since the overall bound needs to hold for all of the above. Then, bounding the resulting failure probability by $\delta'$ gives

$$2SA\ell \exp\left\{-\frac{2m\left([\eta(1-\gamma)^2/2 - \gamma^\ell]/\gamma\right)^2}{(1-\gamma)^{-2}}\right\} \leq \delta'.$$

Solving for $m$ leads to

$$m \geq \log\left(\frac{2\ell SA}{\delta'}\right) \frac{2\gamma^2}{(1-\gamma)^2(\eta(1-\gamma)^2 - 2\gamma^l)^2}$$

and the claim follows by multiplying both sides by $\ell$. $\qquad\square$

*Remark.* Observing that, for fixed $\gamma$, $\ell = O(\log \eta)$, we have $m\ell = O(\eta^{-2}[\log(SA/\delta') + \log(1/\eta) + \log\log(1/\eta)])$, matching the bound of Kearns & Singh (1998, Theorem. 1) up to an additive term $\eta^{-2}\log A$.

### A.1.4  Proof of Lemma 4

Lemma 4 combined with Lemma 3 give the overall sample complexity of the phased-Q learning stage of our algorithm.

**Lemma 4.** *The number of sample transitions from the exploration policy needed to perfectly simulate the parallel sampling model with failure probability less than $\delta''$ is upper bounded by*

$$\frac{2t^{\pi_X}}{\log(2)p_{\min}} \log\left(\frac{2}{p_{\min}}\right) \log\left(\frac{SA}{\delta''}\right).$$

*Proof.* Let $\rho^T$ be the distribution after rolling out the policy for $T$ steps from any starting distribution. By Levin & Peres (2017, Sec. 4.5), we know that $T \geq t^{\pi_X} \log(1/\tau)/\log(2)$ implies $\|\rho^T - \rho^{\pi_X}\|_{TV} \leq \tau$. We choose

$$T \geq t^{\pi_X} \log(2/p_{\min})/\log(2),$$

such that $\tau \leq p_{\min}/2$. Assume that we collect a sample every $T$ steps. After collecting $N$ samples, the probability that a given state-action pair is never observed is bounded from above by $(1 - p_{\min} + \tau)^N \leq \exp(-Np_{\min}/2)$. Bounding this probability by $\delta''$, taking a union bound over all state-action pairs and solving for $N$ yields

$$N \geq 2/p_{\min} \log(SA/\delta'').$$

The claim is obtained by multiplying $N$ and $T$. $\qquad\square$

## A.2  Bounding Total Variation Distance Between Expert and Imitator

This section contains the proof of Theorem 1, as well as the needed auxiliary lemmas.

### A.2.1  Proof of Lemma 5

**Lemma 5** (Relaxation of Theorem 1 from Kakade (2001)). *Suppose policy $\pi$ is $\epsilon$-suboptimal or better at each state with respect to the discounted return. Let $\pi^*$ be the optimal policy under the average reward criterion.*

*Taking $\Sigma$ to be the matrix of right eigenvectors of $P_{\pi^*}$ with corresponding eigenvalues: $\lambda_1 = 1 > ... \geq |\lambda_n|$ Then, under Assumption 4, we have*

$$\mu^\pi \geq \mu^{\pi^*} - \kappa(\Sigma)\|\mathbf{r}\|\frac{1-\gamma}{1-\gamma|\lambda_2|} - (1-\gamma)\epsilon,$$

*where $\kappa(\Sigma) := \|\Sigma\|_2\|\Sigma^{-1}\|_2$ is the condition number of a matrix, and $\mathbf{r}$ is the vector with entries $r(s, \pi^*(s))$ for each state.*

*Proof.* Our proof closely follows that of Kakade (2001). Let $\pi^{\gamma^*}$ be the optimal policy under the discounted return criterion. Then we have, for all $s$

$$V^{\pi^{\gamma^*}}(s) \geq V^{\pi^*}(s).$$

Subtracting $\epsilon$ from both sides gives

$$V^\pi(s) \geq V^{\pi^{\gamma^*}}(s) - \epsilon \geq V^{\pi^*}(s) - \epsilon,$$

for all states, where the first inequality follows from the assumption that $\pi$ is $\epsilon$-suboptimal or better. From here we make use of the relationship between average reward and discounted reward:

$$\begin{aligned}
\mu^\pi &= (1-\gamma)\sum_s \rho_\pi V^\pi(s), \\
&\geq (1-\gamma)\sum_s \rho_\pi(s)(V^{\pi^*}(s) - \epsilon), \\
&\geq (1-\gamma)\sum_s \rho_\pi(s)(V^{\pi^*}(s)) - (1-\gamma)\epsilon, \\
&\geq \mu^{\pi^*} - \kappa(\Sigma)\|\mathbf{r}\|\frac{1-\gamma}{1-\gamma|\lambda_2|} - (1-\gamma)\epsilon.
\end{aligned}$$

The second inequality comes from the fact that $\rho_\pi$ sums to one across states. The final inequality follows exactly from the proof in Kakade (2001). □

### A.2.2 Proof of Theorem 1

First we reproduce two key lemmas from Ciosek (2022) for convenience.

**Lemma 6** (Lemma 5, Ciosek (2022)). *Given an expert dataset consisting of $|D_E|$ points, a policy, $\pi$ which maximises the average intrinsic reward achieves an expected average intrinsic reward of at least*

$$\mu_{int}^\pi := 1 - \nu - \sqrt{\frac{8St^{\pi_E}}{|D_E|}},$$

*for all error terms $\nu > 0$, with probability at least*

$$1 - \delta''' := 1 - 2\exp\left(-\frac{\nu^2|D_E|}{4.5t^{\pi_E}}\right).$$

**Lemma 7** (Lemma 7, Ciosek (2022)). *An agent which achieves an average reward of $1 - \epsilon'$ on the intrinsic reward MDP also achieves average reward of*

$$(1 - \epsilon')\mu^{\pi_E} - 4t^{\pi_E}\epsilon,$$

*on the true MDP.*

Recall that $\mu^\pi$ is the average per-step reward achieved by a policy $\pi$ on the true MDP. Per-step imitation regret with respect to an expert policy $\pi_E$ is then defined as

$$R_{\pi_E}(\pi) = \mu^{\pi_E} - \mu^\pi,$$

and the steady state distribution of a policy, $\pi$, over state, action pairs is given by $\rho_\pi$. $\mu^\pi_{\text{int}}$ gives the average reward obtained on the intrinsic reward MDP by policy $\pi$.

**Theorem 1.** *Consider an instance of algorithm 1. Under assumptions 1-4, in order to achieve regret on the average reward problem of less than $\epsilon$, or total variation distance between the expert and the imitation policy less than $\epsilon$, with probability $1 - \delta$, we need to sample*

$$\max\left\{\frac{1}{\epsilon^2} 32St^{\pi_E}(1 + 4t^{\pi_E})^2, \frac{4.5t^{\pi_E}16(1 + 4t^{\pi_E})^2 \log(4/\delta)}{\epsilon^2}\right\}$$

*transitions from the expert policy, and*

$$\mathcal{O}\left(\frac{t^{\pi_X}}{p_{\min}} \log\left(\frac{1}{p_{\min}}\right) \log^2\left(\frac{SA}{\delta}\right) \log\left(\frac{t^{\pi_E}\beta}{\epsilon(1-\lambda)}\right) \frac{(t^{\pi_E})^8\beta^6}{\epsilon^8(1-\lambda)^6}\right)$$

*from the exploratory policy, where $\beta = \kappa(\Sigma)\|\mathbf{r}\|$, as described in Lemma 5.*

*Proof.* Let

$$1 - \epsilon' := \mathbb{E}_{s,a\sim\rho_\pi}[\hat{r}(s,a)] = \mu^\pi_{\text{int}} \tag{7}$$

represent the average reward achieved by the policy learned on the intrinsic reward problem. We begin by directly evoking Lemma 7. This leads to our overall regret on the true problem, given by

$$\mu^{\pi_E} - \mu^\pi \leq \epsilon'\mu^{\pi_E} + 4t^{\pi_E}\epsilon',$$
$$\leq \epsilon'(1 + 4t^{\pi_E}).$$

We then wish to bound the number of samples needed to achieve $\epsilon'$ small enough such that the overall error obeys

$$\epsilon'(1 + 4t^{\pi_E}) \leq \epsilon, \tag{8}$$

with probability $\delta$.

Lemma 5 gives us the means to decompose $\epsilon'$ in terms of the error incurred by batch learning and construction of the intrinsic reward problem in our algorithm. Let the policy learned by phased Q-learning be $\eta$-suboptimal with respect to the discounted intrinsic return across all states. Lemma 5 allows us to move to the intrinsic reward setting, by bounding the distance between the average intrinsic reward obtained by our learned policy and the optimal intrinsic average reward using this quantity. However the optimal intrinsic average reward itself is a function of the amount of data we have. Lemma 6 bounds this additional error incurred from a potential lack of expert data. Specifically, we define the reward construction error as

$$\epsilon'' := \nu + \sqrt{\frac{8St^{\pi_E}}{|D_E|}}.$$

From Lemma 6, we have $\mu^{\pi^*}_{\text{int}} \geq 1 - \epsilon''$, with some probability $1 - \delta'''$, which we will return to bound, once $\epsilon''$ has been bounded. Substituting $\mu^{\pi^*}_{\text{int}} = 1 - \epsilon''$ and $\epsilon = \eta$ in Lemma 5 we have:

$$\mu^\pi_{\text{int}} \geq 1 - \nu - \sqrt{\frac{8St^{\pi_E}}{|D_E|}} - \kappa(\Sigma)\|\mathbf{r}\|\frac{1 - \gamma}{1 - \gamma|\lambda_2|} - (1 - \gamma)\eta.$$

From the definition of $\epsilon'$ in Equation (7), this leads to the following bound

$$\epsilon' \leq (1 - \gamma)\frac{\kappa(\Sigma)\|\mathbf{r}\|}{1 - \gamma|\lambda_2|} + (1 - \gamma)\eta + \nu + \sqrt{\frac{8St^{\pi_E}}{|D_E|}},$$

which we set to

$$\underbrace{(1-\gamma)\frac{\kappa(\Sigma)\|\mathbf{r}\|}{1-\gamma|\lambda_2|}}_{\text{T1}} + \underbrace{(1-\gamma)\eta}_{\text{T2}} + \underbrace{\nu}_{\text{T3}} + \underbrace{\sqrt{\frac{8St^{\pi_E}}{|D_E|}}}_{\text{T4}} \leq \frac{\epsilon}{(1+4t^{\pi_E})},$$

in order to satisfy Equation (8).

Since there are four terms here with largely independent algorithmic parameters, which we refer to as $\text{T1} - \text{T4}$, we allow error to be distributed evenly across all the terms, which allows us to bound the above expression termwise.

In order to control the first term, we need to set $\gamma$ large enough to satisfy

$$\text{T1} \leq \frac{1}{4}\frac{\epsilon}{(1+4t^{\pi_E})}.$$

For convenience we define $\alpha := 4(1+4t^{\pi_E})/\epsilon$ and $\beta := \kappa(\Sigma)\|\mathbf{r}\|$. This leads us to

$$\frac{(1-\gamma)\beta}{1-\gamma|\lambda_2|} \leq \frac{1}{\alpha}. \tag{9}$$

We set

$$\gamma = \frac{\alpha\beta - 1}{\alpha\beta - |\lambda_2|}, \tag{10}$$

or, equivalently

$$(1-\gamma) = \frac{1-|\lambda_2|}{\alpha\beta - |\lambda_2|}.$$

in order to satisfy Equation (9). Setting $\gamma$ in this way thus allows us to minimise the error incurred by transferring from the discounted to the average reward setting, by increasing the effective horizon of the discounted problem.

Moving on to the second term, directly choose

$$(1-\gamma)\eta = \frac{1}{\alpha},$$

to satisfy $\text{T2} \leq 1/\alpha$. We leave the error in this form, since $\eta$ is multiplied by a factor of $1-\gamma$ in Lemma 3. Intuitively, this corresponds with the scaling of average per step rewards in the discounted problem by a factor of $1/(1-\gamma)$.

Continuing to the terms corresponding to the definition of our intrinsic reward problem, we choose

$$\nu = \frac{1}{\alpha},$$

to satisfy $\text{T3} \leq 1/\alpha$. Finally, we choose

$$|D_E| = \left\lceil \frac{1}{\epsilon^2}128St^{\pi_E}(1+4t^{\pi_E})^2 \right\rceil, \tag{11}$$

where $\lceil \cdot \rceil$ is the ceiling function which chooses the smallest integer greater than the argument. This allows us to satisfy

$$\text{T4} = \sqrt{\frac{8St^{\pi_E}}{|D_E|}} \leq \frac{1}{\alpha}.$$

This provides us with our first bound on the number of expert samples, and completes the bounding of overall error by the relevant factors. To complete our proof, we must ensure that these errors hold with overall probability $\delta$.

With the error of our algorithm controlled, we move on to bounding the failure probability of our algorithm. Let $\delta'''$ be the maximum probability of the expert policy failing to achieve an intrinsic average reward of $1 - \epsilon''$ as in Lemma 6. From Lemma 6, substituting $1/\alpha$ for $\nu$, we have

$$\delta''' \leq 2 \exp \left( -\frac{\left( \frac{\epsilon}{4(1+4t^{\pi_E})} \right)^2 |D_E|}{4.5 t^{\pi_E}} \right).$$

Setting this to be less than $\delta/2$ and rearranging leads to the bound

$$|D_E| \geq \frac{72 t^{\pi_E} (1 + 4t^{\pi_E})^2 \log(4/\delta)}{\epsilon^2}. \tag{12}$$

This is our second bound on $|D_E|$. Taking the maximum over Equation (11) and Equation (12) gives our overall bound on the number of expert samples needed.

To bound the failure probability of phased Q-learning, we first bound the probability that our simulation of the parallel sampler fails. Invoking Lemma 4 with the substitution $\delta'' = \delta/4$, we get

$$N \geq \frac{2t^{\pi_X}}{\log(2)p_{\min}} \log \left( \frac{2}{p_{\min}} \right) \log \left( \frac{SA}{\delta/4} \right). \tag{13}$$

With our remaining error budget, we can look to bound the probability that phased Q learning fails. Substituting $1/\alpha$ for $(1-\gamma)\eta$ and $\delta' = \delta/4$ in Lemma 3 gives

$$m\ell \geq \log \left( \frac{2\ell S A}{\delta/4} \right) \frac{2\gamma^2 \ell}{(1-\gamma)^2 (\frac{1}{\alpha}(1-\gamma) - 2\gamma^\ell)^2}. \tag{14}$$

To complete the proof, we recall from Equation (6) that

$$\ell \geq \log_\gamma \left( \frac{\frac{1}{\alpha}(1-\gamma)}{2} \right). \tag{15}$$

Together, Equations (10) and (12) to (15) give the sample complexity needed to bound the failure probability of our procedure, given by $\delta' + \delta'' + \delta'''$, below the global error $\delta$, with the overall number of expert samples given by the maximum over Equation (11) and Equation (12), and the number of exploratory samples given by $|D_X| \geq N\ell m$.

While the above gives the most precise version of our bound, it is difficult to interpret the effect of changing $\epsilon, \delta$, and the parameters of the environment, largely due to the obscuring influence of $\gamma$ on $m$ and $\ell$.

To fully simplify, we will choose sensible values for $\ell$ and $\gamma$, which will lead to a bound with a simpler form. Since $\gamma < 1$, $\log_\gamma$ is a decreasing function, we choose

$$\ell = \log_\gamma \left( \frac{\frac{1}{\alpha}(1-\gamma)}{4} \right).$$

Substituting this for $\ell$ in Equation (14) leads to the following bound on $m$:

$$m \geq \log \left( \frac{2SA \log_\gamma \left( \frac{\frac{1}{\alpha}(1-\gamma)}{4} \right)}{\delta/4} \right) \frac{2\gamma^2}{(1-\gamma)^2 \left( \frac{(1-\gamma)}{\alpha} - \frac{(1-\gamma)}{2\alpha} \right)^2}$$

$$\geq \log \left( \frac{2SA \log_\gamma \left( \frac{\frac{1}{\alpha}(1-\gamma)}{4} \right)}{\delta/4} \right) \left( \frac{8\gamma^2 \alpha^2}{(1-\gamma)^4} \right). \tag{16}$$

To further simplify, we choose a sensible value for $\gamma$. Noticing that our bound on $\gamma$ based on Equation (9) is increasing in $\alpha\beta$, instead of Equation (10), we choose

$$\gamma = \frac{2\alpha\beta - 1}{2\alpha\beta - |\lambda_2|}.$$

Which, since $2\alpha\beta \geq 1$ and $0 \leq |\lambda_2| \leq 1$, remains in $[0,1)$, and satisfies Equation (9). Substituting this into our choice of $\ell$ and simplifying leads to

$$\ell = \frac{\log\left(\frac{\epsilon^2(1-|\lambda_2|)}{4(1+4t^{\pi_E})(8(1+4t^{\pi_E})\beta-|\lambda_2|\epsilon)}\right)}{\log\left(\frac{8(1+4t^{\pi_E})\beta-\epsilon}{8(1+4t^{\pi_E})\beta-|\lambda_2|\epsilon}\right)}.$$

If we divide $\ell$ by

$$l = \left(\frac{4(1+4t^{\pi_E})\beta}{\epsilon(1-|\lambda_2|)}\right)^2,$$

and take limits, through use of a limit solver, we find that multiple applications of l'Hopital's Rule give

$$\lim_{\alpha\to\infty, \beta\to\infty, \lambda\to 1} \frac{\ell}{l} = 0,$$

which implies that:

$$\ell = \mathcal{O}\left(\left(\frac{4(1+4t^{\pi_E})\beta}{\epsilon(1-|\lambda_2|)}\right)^2\right).$$

All that is left then is to bound the second factor in Equation (16). Substituting our values of $\gamma$, we have

$$\left(\frac{8\gamma^2\alpha^2}{(1-\gamma)^4}\right) = 8\alpha^2 \frac{(2\alpha\beta-1)^2(2\alpha\beta-|\lambda_2|)^2}{(1-|\lambda_2|)^4}$$

$$\leq 8\alpha^2 \frac{(2\alpha\beta)^4}{(1-|\lambda_2|)^4},$$

where the second inequality follows from the fact that $2\alpha\beta \geq 1$. Bringing this together with our first term, this gives an overall bound for $m$ as

$$m = \mathcal{O}\left(\log\left(\frac{SA\left(\frac{t^{\pi_E}\beta}{\epsilon(1-|\lambda|)}\right)}{\delta}\right)\frac{(t^{\pi_E})^6\beta^4}{\epsilon^6(1-\lambda)^4}\right)$$

Multiplying with our bounds for $\ell$ and $N$, we get the overall sample complexity of the exploratory dataset

$$|D_X| = Nm\ell = \mathcal{O}\left(\frac{t^{\pi_X}}{p_{\min}}\log\left(\frac{1}{p_{\min}}\right)\log^2\left(\frac{SA}{\delta}\right)\log\left(\frac{t^{\pi_E}\beta}{\epsilon(1-\lambda)}\right)\frac{(t^{\pi_E})^8\beta^6}{\epsilon^8(1-\lambda)^6}\right).$$

The bound on total variation follows from the well-known identity

$$\mathbb{E}_{\rho_1}[f] - \mathbb{E}_{\rho_2}[f] \leq \epsilon \implies \|\rho_1 - \rho_2\|_{TV} \leq \epsilon,$$

for two discrete distributions, $\rho_1, \rho_2$ and a function $f$ with range $[0,1]$. See e.g. Ciosek (2022) for a proof. Substituting $\rho_{\pi_E}$ for $\rho_1$, $\rho_\pi$ for $\rho_2$, with $f = r$, gives

$$\mu_{\pi_E} - \mu_\pi \leq \epsilon \implies \|\rho_{\pi_E} - \rho_\pi\|_{TV} \leq \epsilon. \tag{17}$$

Choosing $|D_E|$ and $|D_X|$ sufficiently high as described above leads to the left side holding true as discussed. This implies that with the same number of samples, the total variation distance between expert and imitator is less than $\epsilon$, with probability at least $1 - \delta$. $\qquad\square$

## B  Experimental Details for Gridworld

We now proceed to give the details of the gridworld experiments. The environment is a 4x4 gridworld, with no walls and five actions (up, down, left, right, stay-in-place). The agent receives an (extrinsic) reward of 1 when it stays (i.e. executes the last action) in the upper-left square (goal square) and zero otherwise. We use discount of 0.9. The episode terminates after the agent has obtained reward 1, or after a timeout of 11 steps.

The expert demonstration consists of a single trajectory that navigates from the lower-right corner to the upper-left corner along the lower and left sides of the gridworld, then stays in place in the goal state. The first exploration dataset consists of all paths between any two distinct states in the maze. The second exploration dataset is obtained from the first by removing all transitions from the lower-left state and adding 800 transitions that start in the goal square and stay in place.

IT, FR/SQIL and two variants of ORIL as well as three variants of vanilla behavioral cloning have no hyperparameters when applied to this problem. DWBC has two hyperparameters $(\eta, \alpha)$. We tune these parameters on the real environment[3] using grid search with values $\{0.03125, 0.0625, 0.125, 0.25, 0.5\}$ for $\eta$ and values $\{1.5, 3.0, 6.0, 12.0, 24.0, 48.0\}$ for $\alpha$. The tuning is done separately for the first and second exploration datasets. The results of the tuning are $\alpha^\star = 48.0$ and $\eta^\star = 0.125$ for the first exploration dataset and $\alpha^\star = 24.0$ and $\eta^\star = 0.5$ for the second.

## C  Experimental Details for MuJoCo

Here we provide a complete overview of the empirical evaluation of our algorithm and the baselines.

We used the `d4rl` set of benchmarks for all the algorithms. The exploration data came from the `medium-replay-v2` datasets while the expert data came from the `expert-v2` datasets. The algorithms had access to the whole `d4rl` dataset of exploration data (90% of which was used for training and 10% for validation) and 6 episodes of expert data (5 of which were used for training and one for validation). We evaluated on 3 environments: Hopper, HalfCheetah and Walker and report findings as performance profiles aggregated which aggregate 5 runs of each of the 3 tasks as per Agarwal et al. (2021). The shaded area in performance profiles corresponds to stratified bootstrap confidence intervals. Please refer to Agarwal et al. (2021) for a detailed description.

**Hyperparameters**  We used the same learning rate schedule proposed in `nanoGPT` [4], with slightly modified hyperparameters. Training was done for 20000 iterations, with the minimum learning rate set to 0.0001, the maximum learning rate set to 0.001, and the number of iterations the learning rate decays set to 50000. The dropout parameter was set to 0.1. All computation was done in `bfloat16` precision. The parameters of the transformer heads are given below.

| Hyperparameter | Value |
|---|---|
| number of layers | 6 |
| number of heads | 6 |
| embedding size | 384 |

We plan to release the Imitation Transformer source code upon publication.

We also report the results of parameter tuning. We tested two ways of thresholding the classifier to construct our reward: thresholding at a given predicted probability value (see Figure 3) and thresholding so that a given fraction of datapoints is labeled as the expert see Figure 4).

In addition, we performed an ablation where all the rewards were set to either zero, one or one half. Results are in Figure 5.

---

[3]We are against tuning hyperparameters on the real environment in principle, but in this case the baseline algorithm fails to match the performance of our algorithm even when given this advantage.

[4]https://github.com/karpathy/nanoGPT

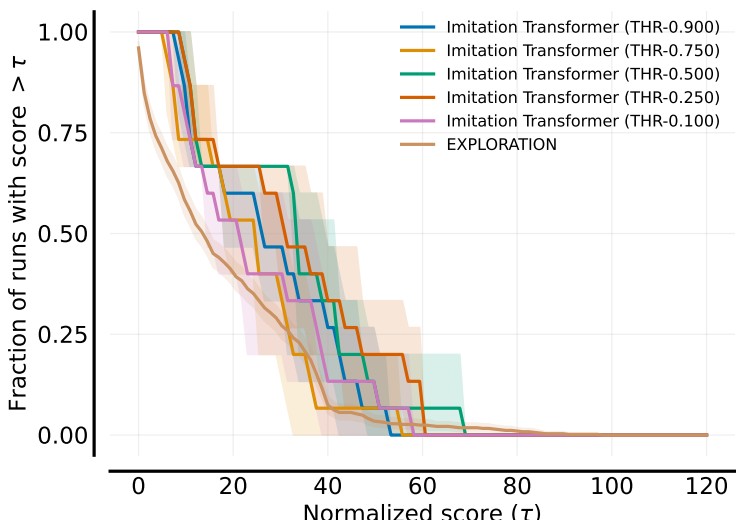

Figure 3: Imitation Transformer performance, thresholding rewards on a fixed probability datapoint comes from expert, for different thresholds.

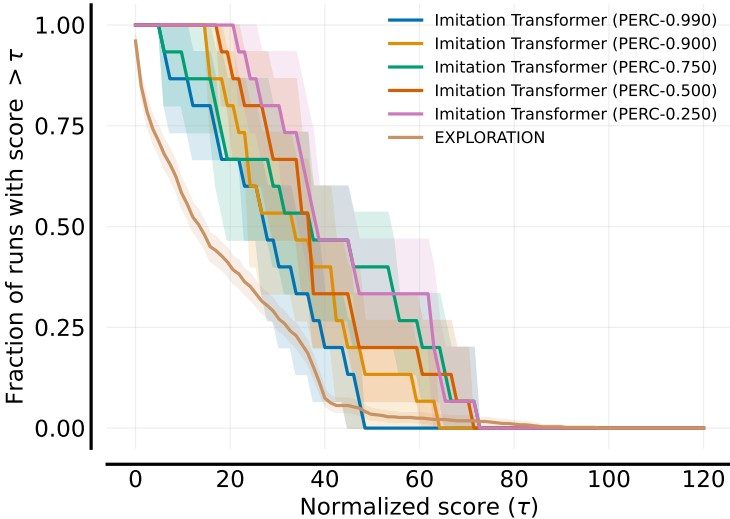

Figure 4: Imitation Transformer performance, thresholding rewards on a probability so that a given fraction of data comes from the expert, for different fractions.

# D   Experimental Details for ATARI

For ATARI, we used the same hyperparameters as for MuJoCo, except the visual observation were pushed through a convolutional network, which converted them to an embedding and was trained end-to-end together with the transformer.

For ATARI, both the expert and the exploration data came from the same public dataset[5]. The dataset was split so that steps between the 10 millionth and 15 millionth index represent the exploration dataset and steps between the 45 millionth and 50 millionth index represent the expert dataset. The remaining part of the dataset was not used.

---

[5]https://console.cloud.google.com/storage/browser/atari-replay-datasets

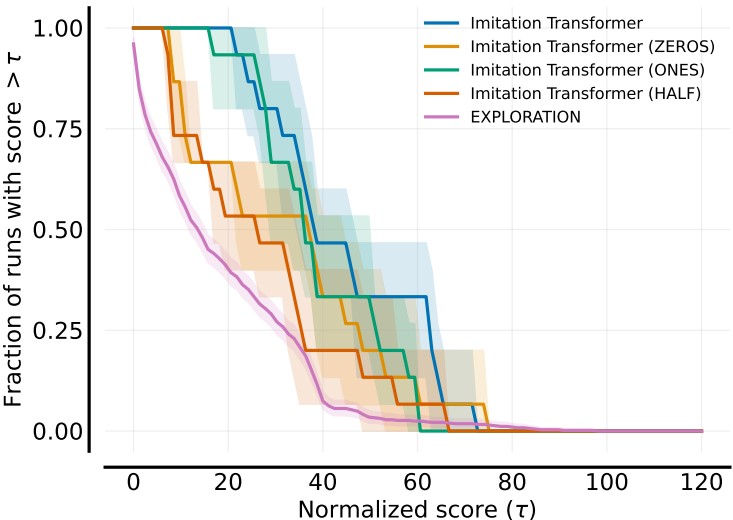

Figure 5: Imitation Transformer performance, ablation for different fixed rewards.

Results of a hyper-parameter sweep aiming to determine the best way to threshold the rewards are given in Figure 6.

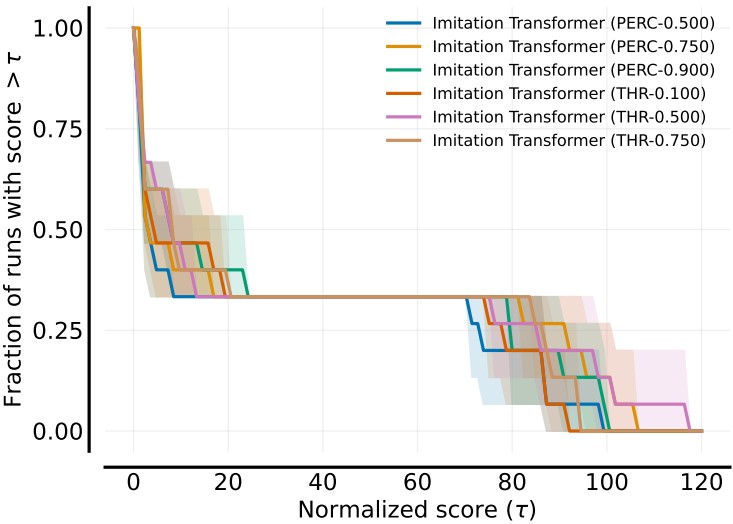

Figure 6: Imitation Transformer performance on ATARI for different ways of thresholding rewards.

