# OpenReview forum: "Imitation Transformer"
_TMLR — Rejected by TMLR_

### Review · Reviewer_X7MX · 2023-12-15

**Summary Of Contributions:**

The paper presents a batch imitation learning method called imitation transformer, and provides the theoretical analysis to back up the idea of the algorithm in an idealised scenario.

**Audience:**

Yes

**Broader Impact Concerns:**

No concerns about the broader impact statement.

**Claims And Evidence:**

No

**Requested Changes:**

On the question below ("Are the claims made in the submission supported by accurate, convincing and clear evidence?") the answer is currently 'No' due to the reasons of the lack of clarity in describing the contributions, but I think it can be resolved if the authors duly address the comments below. The findings are interesting and can surely be of interest to the community, however it is necessary to clarify and contrast how they relate to the state-of-the-art, justify the decision transformer architecture (which takes very modest space in the text despite the title), and clarify why and to what extent does the proposed tabular theoretical setting provide insights into the continuous setting (maybe by comparing them on the identical tabular setting to measure the discrepancy).

Comments, questions and suggestions:

1. A presentation comment on the abstract: I think it could be made more coherent, as it is an excellent chance to make the work of the authors look exciting to the reader. Now it looks like the authors have number of separate seemingly disjoint contributions none of which answer why it advances our knowledge in imitation learning (in essence claiming: "we propose batch imitation method, develop a highly scalable implementation, provide a regret bound, and then perform the experiments"), which they put together into an abstract. It is not a critical issue on its own, but especially in conjunction with the comment about the lack of clarity of the overall paper I think it would greatly improve the first impression if the authors tell instead about the contributions and how it all helps overcome research gap.


2. *“This work addresses batch imitation learning (BIL), an emerging setting which arises at the intersection of two system design constraints.”* Can one call it emergent? I cannot say right now who started to consider the setting of batch imitation learning, but one could trace the term at least back to 2017 [1] and the others have been using similar techniques back to 2006 [2] although not mentioning the name of batch imitation learning.

3. *“However, unlike in these previous works, we do not require online interaction with the environment in order to optimize for the constructed reward.”* I wonder if the authors could contrast the proposed approach with [3]? It considers batch imitation learning in the offline setting.

4. *“However, in order to enable more realistic settings where states can be vectors or images”* Are these settings more realistic, or just different? One might just say that this work considers tabular setting and then shows how to overcome the limitation of finite tabular spaces.

5. Contributions: I would suggest to provide more structure to the section, maybe presenting it as a set of points where the authors list contributions (1), (2), (3)

6. *“Equation (1) — see, e.g. Ciosek (2022)”* Would be important to give a more precise link where to find this equation in Ciosek (2022)

7. It looks like the review of the background is taking up almost 5 pages out of 10 (with the exception of parts of Section 3 and Lemma 1). I wonder if it is possible to make it more concise, perhaps making more of Section 2, especially the discussion around the TV distance, as a part of Appendix or referencing the existing literature, and give more way to clarifying the contributions.

8. (Minor point, not related to the overall assessment) *‘Makes use a recurrence relation’* -> of a recurrence relation

9. What’s the role of the transformer in the paper? It describes that aspect in a very little detail.

10. Analysis is given for the tabular setting and targets , which raises questions about the reality gap between the empirical and theoretical setting.  Could the authors clarify upon it?

[1] Pan, Yunpeng, et al. "Learning deep neural network control policies for agile off-road autonomous driving." The NIPS Deep Rienforcement Learning Symposium. 2017.

[2] Muller, Urs, et al. "Off-road obstacle avoidance through end-to-end learning." Advances in neural information processing systems 18 (2005).

[3] Jarrett, Daniel, Ioana Bica, and Mihaela van der Schaar. "Strictly batch imitation learning by energy-based distribution matching." Advances in Neural Information Processing Systems 33 (2020): 7354-7365.

**Strengths And Weaknesses:**

Strenghths:
- Novelty: The paper has both theoretical and empirical contribution merit, and could be a good fit for the journal subject to addressing the comments
- Soundness: I checked the derivations, and the findings look theoretically and empirically sound

Weaknesses:
- Clarity: it is difficult to follow the text (see comments before) and figure out which parts are original and which ones are following from the previous literature, notably Ciosek (2022); the aspects related to the transformer architecture and its justification are reduced to the minimum despite the title.
-  Soundness: while the theoretical analysis is welcome, it looks overly idealistic, which may not lead to much insight into how the actual proposed method works
- Contribution claims, in relation to clarity: it is difficult to grasp what parts of the paper are actually novel: the original lemmas are intertwined with the ones from Ciosek (2022), the aspect that the paper is using a transformer is not linked to the rest of the paper or motivated in any way; it is not clear also to what extent is the performance  due to decision transformer architecture and to what extent to the original algorithm

---

> ### Author Response · Authors · 2023-12-22
> **Thanks for the review!**
>
> First of all, we wanted to thank the reviewer for taking the time to read the paper in detail and for the high-quality feedback.
>
> We want to begin by addressing the weaknesses. Concerning clarity, we will make it more clear in a revised version of the manuscript which results are new relative to the work of Ciosek (2022) (these are the ones relating to finite exploration data and the batch setting). Concerning the fact that the theoretical analysis is idealistic, we agree with this criticism to an extent but still want to emphasize several benefits of our theoretical contribution. First, one crucial learning from the theory which was previously overlooked is that when one uses a classifier-based method to distinguish expert and demonstration states, one *has* to threshold the classifier, rather than just use the probabilities as rewards as done in ORIL. This learning from theory transfers to the transformer / function approximation where ORIL under-performs our method, despite being intuitively the right thing to do. We will emphasize this learning more in the revised version of the paper. Second, having a finite-data bound (even if in a slightly idealized setting) is better than not having one. For example DemoDICE (Kim et al., 2022) does not come with any sort of finite data guarantee at all. In the batch RL setting, finite-data bounds are perhaps even more important than in standard RL. That's because if we have infinite expert data, we can just do behavioral cloning and if we have infinite exploration data, we can learn a perfect world model and do online RL against it. On the other hand, our analysis, while idealistic and focused on the tabular case, quantifies the performance of the obtained policy under finite expert and exploration datasets. Finally, concerning the usefulness of the transformer architecture, we agree with the reviewer that our methodology can in principle work with any batch RL oracle, not just the decision transformer. We picked the decision transformer because, by reducing batch RL to a supervised learning problem, training becomes much more stable. In a revised version of the paper, we will (as suggested by the reviewer) compare the decision transformer and the ground truth batch RL solution in the tabular setting. We will also provide ablations comparing our batch IL algorithm with decision transformer performance on the true rewards, helping to shed light onto where the performance really comes from.
>
> We now wanted to comment on the requested changes.
>
> Ad 1. We will rewrite the abstract to be more coherent in a revised version of the paper.
>
> Ad 2. We will remove claims that the batch IL setting is emergent. We were not aware of the work of Pan (2017) when submitting the manuscript.
>
> Ad 3. We will contrast our approach with that of Jarrett et al. (2020) in a revised version of the manuscript.
>
> Ad 4. The tabular setting is a special case of vector setting with one-hot vectors and therefore more limited. We will fix the phrasing around this in the way you suggest.
>
> Ad 5. We will add more structure to the contributions section.
>
> Ad 6. The equivalence between bounding total variation and guaranteeing return follows from Lemmas 1 and 2 in Ciosek (2022). We will add a footnote explaining this.
>
> Ad 7. We will move the more standard aspects of the background section to the appendix as suggested.
>
> Ad 8. Thanks for pointing this out, we will fix this.
>
> Ad 9. We picked the decision transformer because, by reducing batch RL to a supervised learning problem, training becomes much more stable. Our core batch RL method will work with other batch RL algorithms as well.
>
> Ad 10. There is definitely a reality gap as you say. However, competing methods like DemoDICE do not currently have finite-data guarantees even in the tabular case. We believe having idealized guarantees is better than having none.
>
> After we receive the other reviews, we will provide an updated version of the manuscript which addresses the outstanding issues as indicated above.
>
> References:
>
> [1] Ciosek Kamil Imitation Learning by Reinforcement Learning
>
> [2] Geon-Hyeong Kim et al. Offline Imitation Learning with Supplementary Imperfect Demonstrations
>
> [3] Pan, Yunpeng el al. Learning deep neural network control policies for agile off-road autonomous driving. The NIPS Deep Reinforcement Learning Symposium. 2017.
>
> [4] Jarrett, Daniel et al. Strictly batch imitation learning by energy-based distribution matching. Advances in Neural Information Processing Systems 33 (2020)

---

### Review · Reviewer_fDbR · 2023-12-24

**Summary Of Contributions:**

The paper proposes Imitation Transformer, an algorithm for Imitation Learning from two datasets: one small *expert dataset* containing expert trajectories and a larger *exploration dataset* containing suboptimal trajectories. The proposed algorithm first learns a hard classifier to distinguish expert state-action pairs (labeled "1") from exploration state-action pairs (labeled "0"), and subsequently applies an offline RL algorithm with this 0-1 reward function.

In the tabular case, analysis is provided assuming the indicator function as classifier and phased Q-learning as the off-policy RL algorithm. In the continuous state-action space, an implementation with a transformer-based classifier and the Decision Transformer as the offline RL algorithm is proposed.

Importantly, practically the method is very similar to ORIL (https://arxiv.org/abs/2011.13885). The differences are more-or-less implementational:

- Imitation Transformer thresholds the classifier to output 0-1 rewards, whereas ORIL uses the soft output of the classifier directly
- Imitation Transformer applies Decision Transformer whereas ORIL uses Critic-Regularized Regression in the Offline RL step

Experiments on Gridworld exploration and on D4RL tasks, indicate that the proposed Imitation Transformer is competitive against ORIL and Behavior Cloning. Additionally, experiments on 3 Atari games show that Imitation Transformer outperforms ORIL but performs similar to Behavior Cloning.

The key novel contribution of the paper seems to be the theoretical analysis. The practical algorithm is too similar to ORIL to consider it to be novel. The only modification which is emphasized by the authors is the hard 0-1 reward. But it is not sufficiently evaluated to say decisively if it is crucial. I provide more reasons for this judgement below.

**Audience:**

Yes

**Broader Impact Concerns:**

Ethical concerns discussed sufficiently

**Claims And Evidence:**

No

**Requested Changes:**

The paper needs significant changes. As I mentioned above, it either needs to emphasize more the theoretical contribution and expand the theory section, or it needs to expand the experiment section. If the theory is expanded, then the theoretical findings should be also illuminated with (toy) experiments.

If the practical algorithm is emphasized, then since it is very close to ORIL, the differences must be ablated more thoroughly, and the claim that IT is better than ORIL should be verified more thoroughly. Some example ablations that can be added

1) How does ORIL perform with 0-1 labels?
2) How does Imitation Transformer (IT) perform with soft labels?
3) Since a key empirical claim is that IT performs better than ORIL, results on the environments used in the ORIL paper should be provided. They used Robotic Manipulation and the DeepMind Control Suite, whereas the present paper evaluates on D4RL. This makes it hard to compare whether the improvement is due to the hard thresholding and Transformer or is it due to using a different environment.

**Strengths And Weaknesses:**

## Strengths
- The theoretical analysis seems novel, though heavily based on Ciosek (2022). As the authors point out, the analysis only applies to tabular MDPs and is using phased Q-learning, which is "somewhat impractical, but its analysis is tractable".
- The authors do provide a practical implementation and show superior performance on D4RL and a tabular exploration environment.
- The presentation is clear and the citations are provided where results from prior work are used.

## Weaknesses
The paper is lacking a focus on a key contribution. If the paper should be more theory-oriented, it needs to provide some illustration of the derived theoretical results. E.g., maybe show that these bounds are obeyed on some toy tasks, or expand the theory in some way. On the other hand, if the paper aims to argue that hard 0-1 labels are better in practice (which sounds a bit counterintuitive because it is erasing information), then it needs a more thorough evaluation of this claim. Namely,

1) How does ORIL perform with 0-1 labels?
2) How does Imitation Transformer (IT) perform with soft labels?
3) Since a key empirical claim is that IT performs better than ORIL, results on the environments used in the ORIL paper should be provided. They used Robotic Manipulation and the DeepMind Control Suite, whereas the present paper evaluates on D4RL. This makes it hard to compare whether the improvement is due to the hard thresholding and Transformer or is it due to using a different environment.

Finally, from the paper it may appear that IT is the best imitation learning algorithm. However, e.g., results of DemoDICE are not provided on D4RL. And furthermore there is a more recent CSIL algorithm (https://arxiv.org/abs/2305.16498) that is even stronger. So, it should be clarified in the paper that IT is not aiming at SOTA and the current best methods should be mentioned.

---

> ### Author Response · Authors · 2023-12-26
> **Thank you for the review!**
>
> Thank you for the review!
>
>
> We first wanted to make a comment about novelty relative to ORIL. Our main contribution lies in providing an algorithm about which we can prove a useful guarantee. Similar guarantees will not exist for ORIL since it fails to solve some gridworld examples (it fails on our second exploration dataset on the gridworld), even if we assume a perfect batch RL oracle. The difference in algorithm specification may seem small, but the difference in the outcomes is large. We will make this clearer in a revised version of the paper.
>
>
> We wanted to address the questions you bring up.
>
>
> Ad 1 / 2. In our experiments, we wanted to compare like with like. This means that in our implementation of ORIL, we also used the decision transformer rather than the batch RL method used by the original ORIL paper. This allows us to isolate the effect of thresholding. We believe our current experiments show thresholding is (somehow counterintuitively) needed in practice since the presence of thresholding is the only difference between our version of ORIL and IT. We agree this is not explained well and will adjust the wording about this in a revised version of the paper.
>
>
> Ad 3. We agree that more experiments would make the paper stronger, but at the same time, we have limited time and computational resources. Our goal is not to provide a state-of-the-art method evaluated on every benchmark, but instead to (1) provide a finite-data guarantee in the tabular case (which many other methods lack) and (2) see how the learnings from theory, specifically the importance of thresholding, transfer to the function approximation case. We believe our experiments are mostly sufficient to support these contributions. However, we will add more experiments to illustrate the theory in a revised version of the paper.
>
>
> Once we receive all reviews, we will provide a revised version of the paper which addresses the points above. We will also provide a comment to all reviewers outlining the new framing and the contributions of the revised version of the paper.

---

### Review · Reviewer_eWv5 · 2024-01-28

**Summary Of Contributions:**

This paper proposes an imitation learning method that works well in the setting where an expert dataset and a (potentially larger) exploration dataset is provided. Compared to previous works in this direction, the authors propose to use an offline RL algorithm instead of having unlimited interaction with the environment to learn from the datasets, once an intrinsic reward is assigned to the data points based on whether they come from the expert. The authors provide theoretical results to support the use of the new method and understand its behavior in tabular setting with respect to different settings of the MDP, the expert dataset, and exploration dataset. The authors also present empirical results show that the proposed method can outperform some previous methods in a gridworld example and also a few tasks in MuJoCo and Atari.

**Audience:**

Yes

**Claims And Evidence:**

Yes

**Requested Changes:**

**Critical for recommendation:**
- I feel there needs to be a better/more extended motivation discussion on what are the settings and tasks that the proposed method can have a real advantage over the many other offline learning algorithms that have been studied in the past few years. I am not sure if the discussed MuJoCo/Atari tasks are the best for showcase this since they originally do have reward signals.

**Strengthen the work:**
- Can be good to test on more tasks and settings but this is not the most important.

**Strengths And Weaknesses:**

**Strengths**:
- interesting theoretical results
- experiments in toy examples as well as high-dimensional tasks show improved performance
- some effort is made to ensure the comparisons are statistically significant
- overall clear presentation and writing

**Weaknesses**:
- significance: it is a bit unclear to me how does the proposed method compare to other imitation learning that utilize reward in one way or another, (for example, those that perform some form of weighted behavioral cloning with the weight proportional to the reward or closeness to the expert policy), and for the transformer experiments, if compared to decision transformer in the original setting (with rewards), does the proposed method have an advantage? Or is the proposed method only good in settings where a reward is not given?
- empirical results: the empirical results are obtained from a limited set of dataset/tasks, and in some results, for example the Atari ones the performance of the proposed method is essentially the same as some basic BC variants.
- motivation: it is not very clear to me if there are practical settings where the proposed method can be really good compared to other offline learning techniques, or perhaps the MuJoCo and Atari environments are not exactly the best cases to showcase its advantages.
- meaningful comparison: I don’t think in the ORIL paper they experimented on the grid world or MuJoCo/Atari in the same setting. Have you finetuned ORIL to your settings to make sure the baseline is  a strong baseline?

---

> ### Author Response · Authors · 2024-02-14
> **Thanks for the review (1/2).**
>
> Thank you for your feedback.
>
> We now wanted to comment on the weaknesses identified in your review.
>
> 1. Regarding your remarks on significance, we want to emphasize that our method addresses the setting where reward signal is not available, therefore we think that comparing it to methods that make use of rewards is not very useful. Of course, a method that has access to demonstrations *and* rewards can be expected to do better than a method that only has access to the demonstrations but not to rewards, because it has strictly more information.
> Our setting (without access to rewards) is of practical significance wherever rewards are hard to specify or where we fear reward hacking might be an issue. Having said that, we agree it may be useful to consider an ablation where we compare our method to a decision transformer with an identical architecture but additional access to the true rewards. We will include those additional results in a revised version of the paper. Concerning variants of weighted behavioral cloning, we already compare to discriminator-weighted behavioral cloning, which, like our method, does not depend on the reward signal and find our method to have superior performance. Concerning the additional baselines you suggest (behavioral cloning with weights computed form the reward or distance from expert data), we are concerned that these approaches are heuristics that would only work well for very specific MDPs. This is because planning for multiple steps ahead is necessary to solve MDPs well and weighted behavioral cloning does not typically do this (unless transition data is used to compute the weights).
>
> 2. We claim that our experimental evaluation is modest but sufficient. We did not have the computational resources to do a more extensive analysis. We did not aim to provide a state-of-the-art method evaluated on every benchmark, but instead to (1) provide a finite-data guarantee in the tabular case (which many other methods lack) and (2) see how the learnings from theory, specifically the importance of thresholding, transfer to the function approximation case. We believe our experiments are sufficient to support these contributions. However, we will add more experiments to illustrate the theory in a revised version of the paper. Concerning the fact that our method does not outperform Behavioral Cloning on ATARI, we wanted to emphasize two things. First, we believe there is value in reporting all relevant research findings to the community, including experiments that did not produce an significant improvement in performance. Second, our method is not substantially more complex to implement than behavioral cloning (it performs supervised learning twice), while working much better on MuJoCo and the gridworld benchmarks. We claim this still makes it preferable to behavioral cloning in practice in certain scenarios.
>
> 3. There are many practical tasks where reward is unavailable. Self-driving is a classic example, creation of human-like experiences for computer games is another. Such problems have been studied extensively in the context of on-line imitation learning and are well-established. Our work attempts to combine the benefits of Imitation Learning with the idea of learning from an offline dataset, without a simulator. Moreover, we believe that experimentally evaluating on MuJoCo, Atari and the gridworld is sufficient to study the proposed method at a level typical of a good TMLR publication. There are many papers in comparable venues on related topics that employ these or similar benchmarks. Of course solving MuJoCo and ATARI is not the ultimate goal. These benchmarks are used because they come with convenient datasets / simulators, while exposing the agent to a diverse enough set of challenges to make the comparison worthwhile. Once a method is validated using these benchmarks it can be deployed in more elaborate or more practical settings. However, we claim that this further step is beyond the scope of a typical TMLR paper.
>
> 4. We believe in the importance of comparing like-for-like to obtain meaningful results. Specifically, for ORIL this means that: (1) we used Q-learning batch RL solver for the gridworld (which is the same that was used for our method) and (2) we used the Decision Transformer batch RL solver for ORIL for MuJoCo and ATARI (which is again the same that was used for our method). Indeed, the presence of reward thresholding is the only difference between the implementations of our version of Imitation Transformer and ORIL. This means we do have a meaningful comparison.

---

> ### Author Response · Authors · 2024-02-14
> **Thanks for the review (2/2).**
>
> Concerning the requested changes, we claim that MuJoCo and ATARI are sufficient benchmarks to study imitation learning and indeed have been used in many imitation learning papers. The fact that the simulators also provide a reward is an advantage, not a problem. This is because, while we do not use the reward for learning the agent, we do use it as an unambiguous metric for post-training evaluation, which is very helpful. There are many examples of RL works which use variants of existing benchmarks while providing or withholding certain information from the agent in order to zero in on particular ways of learning the policy. We view benchmarks as tools for scientific inquiry that do not have to be used verbatim but should be adapted to the problem at hand. Also, see our reply to point 3 above.
>
> We will now work on a new, improved version of the paper. Once done, we will also provide a comment to all reviewers outlining the new framing.

---

> > ### Comment · Reviewer_eWv5 · 2024-02-25
> > **Thank you for the rebuttal**
> >
> > I thank the authors for the response, I agree that the particular setting where we do not have reward can be an important setting, and it might not make sense to compare with some other works that utilize rewards.
> >
> > Overall the paper provides some interesting theoretical and empirical results. However I am still a bit not sure about the significance of the results. It seems to me that currently this is a borderline paper. (It will help if the performance is stronger or more analysis is provided to help us understand where the proposed method is highly advantageous. Not necessarily a larger scale experiment.)

---

### Decision · Action_Editor_B2zQ · 2024-03-28

**Recommendation:** Reject

**Comment:**

The proofs wouldn't be independently of interest to the TMLR audience and the empirical results lack some reasonable baselines and often have unclear advantages to the baselines that are present. And the framing of the paper around the transformer architecture feels unjustified.

I think with more clear cut empirical wins and a renaming, this paper could be an interesting contribution, but I'm afraid I must reject it in its current state.

**Audience:**

As previously mentioned, the proofs appear sound, but several reviewers suggest they won't be of interest to the TMLR audience given their similarity to existing work. And while I appreciate the simplicity of the proposed method, the advantages of this approach aren't clear, particularly in light of missing comparisons highlighted by some reviewers.

I understand that papers that combine theory and practice are often greater than the sum of their parts, but I'm afraid the authors have yet to pass the threshold where the results would be of sufficient interest to the TMLR audience.

**Claims And Evidence:**

While I haven't checked the proofs myself, the reviewers all agree that they seem sound. Similarly, the statistical methodology behind the empirical results seems sound (performance profiles are always appreciated).

However, the empirical results (particularly Figure 2) don't provide much evidence for the claim that the "Imitation Transformer" is significantly better than existing approaches.

There is also an issue with the implicit claim of the title: its very unclear how the transformer architecture has any bearing on the methodology proposed.

**Resubmission Of Major Revision:**

The authors may consider submitting a major revision at a later time.